# Accuracy of age estimation and assessment of the 18-year threshold based on second and third molar maturity in Koreans and Japanese

**Sehyun Oh[1], Akiko Kumagai[2], Sin-Young Kim[3], Sang-Seob Lee[1] \***

1 Department of Anatomy Catholic Institute of Applied Anatomy, College of Medicine, The Catholic University of Korea, Seoul, Republic of Korea, 2 Division of Forensic Odontology and Disaster Oral Medicine, Department of Forensic Science, Iwate Medical University, Morioka, Iwate, Japan, 3 Department of Conservative Dentistry, Seoul St. Mary's Hospital, College of Medicine, The Catholic University of Korea, Seoul, Republic of Korea

\* sslee1418@gmail.com

**Data Availability Statement:** All relevant data are within the manuscript and its Supporting information files.

## Abstract

This study aimed to validate Lee's age estimation method and assess the 18-year threshold in Korean and Japanese populations. We evaluated the maxillary and mandibular second (M2) and third molars (M3) in 2657 orthopantomograms of the Korean and Japanese populations aged 15–23 years (19.47±2.62 years for Koreans, 19.31±2.60 years for Japanese), using Demirjian's criteria. Dental age was estimated, and correlations between chronological and dental ages were analyzed. Classification performance was calculated based on the 18-year threshold. The relationship between developmental stage and chronologic age was analyzed using multiple linear regression. Our results revealed that Lee's method was appropriate for estimation in the Korean population. When the Lee's method was applied to the Japanese population, a lower value of correlation coefficients between estimated and chronological age, and lower specificity were observed. Population differences were observed predominantly in the stages of root development (stages F and G) of M2s and M3s in both jaws and more frequently in females than in males. In the multiple linear regression between developmental stage and chronological age, lower values of adjusted $r^2$ were observed in the Japanese population than in the Koreans. In conclusion, the Lee's method derived from the Korean population data might be unsuitable for Japanese juveniles and adolescents. To support the findings of this study, future studies with samples from multiple institutions should be conducted. Future studies with larger sample sizes are also warranted to improve the accuracy of dental age estimation and confirm the developmental pattern of teeth in the Japanese population.

## Introduction

Age estimation is an essential process for human identification. In particular, dental age estimation has been reported to be more accurate than estimation based on other age indicators of the body; hence, it is widely used in forensic practice [1–3]. Various dental age estimation

**Funding:** This research was supported by Catholic Medical Center Research Foundation (SSL, 5-2021-B0001-00309) funded by The Catholic University of Korea and this work was also supported by National Forensic Service (SSL, NFS2022MED08), Ministry of the Interior and Safety, Republic of Korea. The funders had no role in study design, data collection and analysis, decision to publish, or preparation of the manuscript.

**Competing interests:** The authors have declared that no competing interests exist.

methods have been reported, including three types based on the age span of subjects: children, adolescents and young adults, and older adults. In particular, dental age estimation for adolescents and young adults is challenging because most tooth development is completed, and regressive changes due to increasing age have not yet appeared [4]. For this age span, evaluation of the third molars (M3s) is widely used as a method for age estimation because these are the only teeth that are still developing in human dentition. Dental age estimation using M3 has been used extensively. For instance, Mincer et al. [5] evaluated M3 in American Caucasians with stage scheme by Demirjian et al. [6]. The study reported the calculation of the empirical probability of being over 18 years old based on the developmental stage of M3 and predicted chronological age using regression analysis. Solari and Abramovitch [7] evaluated M3 of Hispanics using a 10-stage tooth development scoring system by adding F1 and G1 stages to achieve higher accuracy of the development toward apexification. They concluded the possible application of M3 in estimating age of Hispanics. Gunst et al. [8] classified M3 of Belgian Caucasians using the 10-stage criteria suggested by Gleiser and Hunt [9] and modified by Köhler et al. [10], evaluated the correlation of chronological age with developmental stages of M3, and optimized this approach with a multiple regression model. Additionally, Thevissen et al. [11] evaluated M3 of young adults from Thailand using the same criteria with the study of Gunst et al. [8]. To increase objectivity, the developmental stages of the second molar (M2) were combined to obtain the M3/M2 ratio, and multiple regression was performed to estimate age [11].

Lee et al. [4] evaluated the development of M2 and M3 on orthopantomography in Korean individuals using Demirjian's criteria [6]. Due to the variability in M3, they combined the developmental stages of M2 and M3 to improve accuracy and presented a regression formula for the Korean population. They suggested various regression models using the developmental stage of a single tooth (M2 or M3), using stages of two teeth among M2s and M3s, and using stages of all M2s and M3s in both jaws, as variables. In addition, they treated variables as discrete or continuous, and the results of both types of variables were compared. The Lee's method, after publication, was used in some cases for forensic age estimation in Korea, and some forensic practitioners demanded validation of the method using another Korean population data to prove the accuracy. Based on the Korean Civil Act and Juvenile Act, people aged over 18 years are considered adults [12, 13]. Therefore, when the machinery of law, such as the courts, requested forensic examinations for discrimination of minor/adult status, they requested results based on the 18-year threshold. However, Lee et al. [4] did not determine adult probabilities based on the age of 18 years. For this reason, the study for assessment of the 18-year threshold based on the Korean population data became necessary.

Numerous studies have reported that different populations exhibit distinct timing and rates of M3 development [6–8, 11]. This constitutes a key component of developing a reliable database of the developmental stages of M3 for each population group to increase the accuracy of age estimation. However, some studies have reported that the differences in tooth development between populations are insignificant [14–16]. Korea and Japan are neighboring countries, and genetic similarities between these two populations have been reported in genetic mapping studies of Asian populations [17]. However, there was also a report presenting statistically significant differences in the development of the third molars between the Korean and the Japanese population [18]. Since these contrasting reports existed, it was necessary to verify whether the age estimation method based on the Korean population data is applicable to the Japanese population as well.

This study aimed to validate Lee's method for age estimation in Korean and Japanese populations by evaluating the accuracy of the estimated age. In addition, we performed statistical

analyses to assess the 18-year threshold for both populations for practical use in various forensic contexts.

## Materials and methods

### Sample collection

The sample size was estimated based on multiple linear regression considering four predictors, with a confidence level of 95%, and a statistical power of 80. Supposing the effect size $f^2$ was small (i.e., $f^2$ = 0.03), the minimum sample size was calculated as 403 using the G*Power software (Dusseldorf, Germany). Data were randomly selected and stratified by gender and population from the radiographs which were enrolled from Seoul St. Mary's Hospital, Catholic University of Korea (900 males and 900 females, 19.47±2.62 years), and Iwate Medical University of Japan (406 males and 451 females, 19.31±2.60 years). We used dental panoramic radiographs were taken for the purpose of clinical treatment, aged between 15 and 23 years, comprising retrospective medical records that were reviewed. Radiographs depicting at least one M2 and M3 in both jaws were selected. Radiographs showing severe tooth decay, endodontic treatment, pathological lesions in the jawbone, agenesia, and unclear images were excluded. Radiographs belonging to patients with a medical history of endocrinal disorders were also excluded. The chronological age was calculated as the difference between the date of the radiography and the date of birth [19–21]. The study group was divided into nine age groups by year of chronological age. Table 1 shows the age and sex distribution of the study population. The number of subjects assigned to each stage was distributed similarly (around one hundred for Koreans and fifty for Japanese) to reduce the bias error in the statistical analysis. This study was conducted after an approval from the Institutional Review Board (IRB) of the Seoul St. Mary's Hospital, the Catholic University of Korea (approval no.: KC21WISI0382) and the Ethics Committee of Iwate Medical University, School of Dentistry (approval no.: 01352). Because we cannot obtain informed consent from each of them, the collecting informed consent was waived by IRB. All collected data were anonymized, except sex, date of birth, and the date of taking radiographs. An extreme care was taken to protect leakage of personal information during all phases of the study.

### Estimation of dental age

Developmental stages of M2s and M3s in both jaws were evaluated with the eight-stage criteria presented by Demirjian et al. [6]. For statistical analysis, the tooth from one side of each jaw

**Table 1. Distribution of age and sex of the samples.**

| Age group (years) | Korean | | | Japanese | | |
|---|---|---|---|---|---|---|
| | Male | Female | Total | Male | Female | Total |
| 15 | 100 | 100 | 200 | 48 | 65 | 113 |
| 16 | 100 | 100 | 200 | 44 | 61 | 105 |
| 17 | 100 | 100 | 200 | 42 | 47 | 89 |
| 18 | 100 | 100 | 200 | 45 | 57 | 102 |
| 19 | 100 | 100 | 200 | 47 | 47 | 94 |
| 20 | 100 | 100 | 200 | 46 | 43 | 89 |
| 21 | 100 | 100 | 200 | 45 | 45 | 90 |
| 22 | 100 | 100 | 200 | 45 | 43 | 88 |
| 23 | 100 | 100 | 200 | 44 | 43 | 87 |
| Total | 900 | 900 | 1800 | 406 | 451 | 857 |

was selected to avoid unnecessary duplication. If teeth on each side in the homologue presented with different developmental stages, the lower-stage tooth was selected. Given the uncertainty in prediction of age estimation, the lowest limit of the estimated age should be selected as a conservative measure [22]. Two observers independently evaluated the radiographs after the pre-calibration scoring process. Both observers were well-trained forensic odontologists with experience in forensic examination for more than ten years. One hundred randomly selected orthopantomograms were re-examined three months after the first examinations by the main and the second observer to assess intra- and interobserver reliabilities.

The dental ages of radiographs from Korean and Japanese population were estimated by Lee's methods [4]. Lee et al. performed two different regressions with the variables treating as continuous and discrete data and presented the multiple regression formulae (the result from continuous data) and tables (from discrete data) for age estimation. In addition, they suggested the estimation methods for single tooth, for four combinations of two teeth among M2s and M3s, and for combination of whole M2s and M3s, respectively. We estimated the dental ages of the samples using all possible eighteen methods proposed by Lee et al. [4].

## Statistical analysis

Intra- and interobserver reliabilities for the evaluation of maturity in M2s and M3s, were calculated using Cohen's kappa statistics [23]. Before statistical analysis, the assumption of normality was checked based on the values of skewness and kurtosis and the Shapiro-Wilk test. All continuous variables had univariate normality, as shown by skew values $< 2$ and kurtosis values $< 7$. Descriptive analysis was presented as the mean and standard deviation for the developmental stage of teeth. Due to both estimated and chronological ages satisfied the normality assumption, we tested the Pearson correlation coefficients (PCCs) for observing correlations between the chronological and estimated ages. Agreement tests were assessed by Intraclass correlation coefficients (ICCs) by a two-way random-effects model with an absolute agreement. In addition, Bland-Altman plots were generated for a visual representation of estimated and chronological ages. To assess the 18-year threshold, sensitivity (the ratio of samples that were equal to or over the age threshold and were estimated to being equal to or over the age threshold), specificity (the ratio of samples that were truly under the age threshold and were estimated to being under the threshold), positive predictive value (PPV, the ratio of samples that were estimated to being equal to or over the age threshold and were truly equal to or over the age threshold), negative predictive value (NPV, the ratio of samples that were estimated to being under the threshold and were truly under the age threshold), and accuracy (the ratio of samples that were correctly estimated) were calculated. Population differences in tooth developmental status between Koreans and Japanese were tested using a two-tailed independent t-test. Linear regression was performed to model the relationship between treating the developmental stage of each tooth and chronological age as dependent variables; a residual plot was used to test the quality of the model. Regression was performed using two approaches, that is, treating the developmental stage of each tooth as discrete and continuous variables, as per Lee's suggestions [4]. A two-tailed $P$-value $< 0.05$ was considered statistically significant. All statistical analyses were performed using SAS version 9.4 (SAS Institute, Cary, NC).

## Results

### Datasets

The mean chronological ages of the study populations were 19.47±2.62 years and 19.31±2.60 years for Koreans and Japanese (1800 Koreans and 857 Japanese), respectively. The mean chronological ages for males and females were 19.45±2.64 years and 19.49±2.59 years in the

Korean population, and 19.48±2.60 years and 19.16±2.60 years in the Japanese population. The minimum and maximum ages of the Koreans included in the study populations were 15.00 years and 23.98 years in males, and 15.01 years and 23.97 years in females, respectively. For the Japanese, they were 15.01 years and 23.97 years in males, and 15.00 years and 23.99 years in females, respectively. No significant difference was observed in the chronological ages of females and males in the Korean and Japanese populations ($P > 0.05$).

## Observer reliabilities

The result of kappa for the intra-observer reliability was 0.949 (0.932 to 0.967, 95% confidence interval (CI)), while kappa for the inter-observer reliability was 0.942 (0.923 to 0.961, 95% CI). Both reliabilities were interpreted as "almost perfect" agreement according to the standard by Landis and Koch [24].

## Descriptive statistics of M2s and M3s

The mean and standard deviation of chronologic age according to the developmental stages of M2 and M3 in both populations are presented in Tables 2 and 3. We observed a difference between Korean and Japanese females according to the developmental stages of M2s and M3s (Fig 1). The statistically significant differences were observed between Korean and Japanese males at stage F of maxillary M3 and stage G of mandibular M2 and between Korean and Japanese females at stage G of M2s in both jaws; stage D, F, and G of maxillary M3; and stage C, F, and G of mandibular M3 ($P < 0.01$). Stage H represents the completion of tooth development. Since there is no lack of natural upper limit for the age of stage H in arraying data, the descriptive statistics for this stage is changed according to a used cut-off upper age [4, 25]. Hence, we did not compare the data of stage H between Korean and Japanese population.

## Accuracy of estimated ages

The estimated ages by Lee's method using the data of whole M2s and M3s in both jaws, was compared with chronologic ages in Fig 2. The box plots for the estimated ages with single tooth or combinations of two teeth were showed in S1 Fig. The interquartile ranges of estimated age were broader for the Japanese than that of the Koreans (Fig 2, S1 Table) and this tendency was observed more apparently in age groups under 18 years of age.

The PCCs and ICCs revealed a strong correlation between estimated and chronological ages in the Korean population ($P < 0.001$). The PCCs and ICCs were lower for the Japanese population than for the Korean population (Table 4), and the lowest values were observed for Japanese female samples (0.72 and 0.81, respectively). The difference between chronological and estimated ages is represented visually in a Bland–Altman plots (Fig 3). In the Bland–Altman plots, the Korean population data exhibited a more homogeneous distribution of datapoints around the x-axis than the Japanese population. Moreover, the Korean population data had a narrower confidence interval than the Japanese population, and the datapoints were more densely clustered compared to the Japanese population data.

## Assessment of 18-year threshold

To demonstrate the performance of the 18-year threshold classification, the sensitivity, specificity, PPV, NPV, and accuracy were calculated (Table 5). The sensitivities were 97.0–98.3% for the Korean population and 85.3–95.2% for the Japanese population, whereas the specificities were 94.7–98.0% for the Korean population and 64.2–68.2% for the Japanese population (Table 5).

**Table 2. Comparison of chronological age according to each stage of M2 between the Korean and Japanese populations.**

| | Stg | UM2 | | | | | | P |
|---|---|---|---|---|---|---|---|---|
| | | Korean | | | Japanese | | | |
| | | n | Mean | SD | n | Mean | SD | |
| Male | B (2) | | | | | | | |
| | C (3) | | | | | | | |
| | D (4) | | | | 1 | 16.08 | | |
| | E (5) | | | | | | | |
| | F (6) | | | | 8 | 16.19 | 1.23 | |
| | G (7) | 290 | 16.44 | 1.10 | 97 | 16.76 | 1.51 | 0.053 |
| | H (8) | 610 | 20.88 | 1.83 | 300 | 20.46 | 2.17 | 0.004* |
| Female | B (2) | | | | | | | |
| | C (3) | | | | | | | |
| | D (4) | | | | | | | |
| | E (5) | | | | | | | |
| | F (6) | 2 | 15.44 | 0.61 | 7 | 16.41 | 1.26 | 0.346 |
| | G (7) | 276 | 16.52 | 0.99 | 139 | 17.39 | 1.98 | <0.001** |
| | H (8) | 622 | 20.81 | 1.89 | 305 | 20.03 | 2.41 | <0.001** |
| | Stg | LM2 | | | | | | P |
| | | Korean | | | Japanese | | | |
| | | n | Mean | SD | n | Mean | SD | |
| Male | B (2) | | | | | | | |
| | C (3) | | | | | | | |
| | D (4) | | | | | | | |
| | E (5) | | | | | | | |
| | F (6) | 4 | 15.41 | 0.22 | 2 | 15.56 | 0.30 | 0.502 |
| | G (7) | 291 | 16.58 | 1.18 | 110 | 17.12 | 1.86 | 0.005* |
| | H (8) | 605 | 20.86 | 1.92 | 294 | 20.39 | 2.24 | 0.002* |
| Female | B (2) | | | | | | | |
| | C (3) | | | | | | | |
| | D (4) | | | | | | | |
| | E (5) | | | | | | | |
| | F (6) | 4 | 15.25 | 0.42 | 5 | 16.03 | 0.56 | 0.055 |
| | G (7) | 304 | 16.61 | 0.97 | 128 | 17.27 | 1.92 | <0.001** |
| | H (8) | 592 | 20.99 | 1.77 | 318 | 19.97 | 2.42 | <0.001** |

U, maxilla; L, mandible; Stg, stage; P, P-value; SD, standard deviation. Stages 1 to 8 indicate the scores of Demirjian developmental stages A to H.

*P < 0.01 and **P < 0.001 indicate a statistically significant difference between Korean and Japanese population data.

### Relationships between dental maturity and chronological ages

The relationships between chronological age and degree of mineralization of M2 and M3 were analyzed using multiple linear regression, and the results are presented in Table 6. The relationships between chronological age and developmental degrees of single tooth and two combined teeth are presented in S3 Table for Koreans and S4 Table for Japanese. Adjusted $r^2$ was calculated using multiple regression (Table 6). Regression analysis of the Korean population performed using four teeth based on discrete data revealed the highest adjusted $r^2$ values of 0.834 and 0.855 for males and females, respectively. The coefficients of determination of the Japanese population were 0.693 and 0.596 for males and females, respectively, which were lower than those for the Korean population data. Regression analysis of M3s in both jaws in

**Table 3. Comparison of chronological age according to each stage of M3 between the Korean and Japanese populations.**

| | Stg | UM3 | | | | | | P |
| | | Korean | | | Japanese | | | |
| | | n | Mean | SD | n | Mean | SD | |
|---|---|---|---|---|---|---|---|---|
| Male | B (2) | 1 | 16.89 | | | | | |
| | C (3) | 6 | 15.61 | 0.71 | 4 | 15.94 | 0.81 | 0.511 |
| | D (4) | 60 | 15.69 | 0.86 | 21 | 15.94 | 0.76 | 0.242 |
| | E (5) | 149 | 16.86 | 1.42 | 54 | 16.90 | 1.53 | 0.872 |
| | F (6) | 200 | 18.28 | 1.73 | 69 | 17.52 | 1.65 | 0.002* |
| | G (7) | 156 | 19.60 | 1.56 | 102 | 19.67 | 1.72 | 0.752 |
| | H (8) | 328 | 22.04 | 1.29 | 156 | 21.69 | 1.62 | 0.021* |
| Female | B (2) | | | | 2 | 15.88 | 0.79 | |
| | C (3) | 10 | 15.54 | 0.42 | 10 | 16.07 | 0.97 | 0.140 |
| | D (4) | 72 | 16.08 | 1.01 | 53 | 17.03 | 1.91 | 0.002* |
| | E (5) | 203 | 17.22 | 1.45 | 79 | 17.07 | 1.60 | 0.422 |
| | F (6) | 269 | 19.26 | 1.67 | 89 | 18.60 | 1.99 | 0.006* |
| | G (7) | 136 | 20.67 | 1.64 | 113 | 19.96 | 1.93 | 0.002* |
| | H (8) | 210 | 22.55 | 1.05 | 105 | 21.80 | 1.79 | <0.001** |

| | Stg | LM3 | | | | | | P |
| | | Korean | | | Japanese | | | |
| | | n | Mean | SD | n | Mean | SD | |
|---|---|---|---|---|---|---|---|---|
| Male | B (2) | | | | 1 | 15.04 | | |
| | C (3) | 15 | 15.59 | 0.84 | 6 | 15.87 | 0.62 | 0.471 |
| | D (4) | 57 | 15.77 | 0.99 | 28 | 16.17 | 0.95 | 0.085 |
| | E (5) | 114 | 16.70 | 1.46 | 50 | 16.68 | 1.50 | 0.921 |
| | F (6) | 214 | 17.91 | 1.41 | 65 | 17.77 | 1.58 | 0.488 |
| | G (7) | 216 | 20.07 | 1.62 | 119 | 19.81 | 1.69 | 0.170 |
| | H (8) | 284 | 22.19 | 1.21 | 137 | 21.90 | 1.48 | 0.054 |
| Female | B (2) | 2 | 15.35 | 0.48 | 3 | 15.12 | 0.17 | 0.479 |
| | C (3) | 12 | 15.65 | 0.52 | 29 | 16.50 | 1.23 | 0.004* |
| | D (4) | 114 | 16.40 | 1.20 | 65 | 16.83 | 1.57 | 0.060 |
| | E (5) | 150 | 17.59 | 1.70 | 71 | 17.71 | 1.80 | 0.650 |
| | F (6) | 242 | 18.88 | 1.67 | 66 | 18.24 | 1.87 | 0.007* |
| | G (7) | 222 | 20.95 | 1.63 | 130 | 20.30 | 1.86 | 0.001* |
| | H (8) | 158 | 22.72 | 0.96 | 87 | 22.13 | 1.48 | 0.001* |

U, maxilla; L, mandible; Stg, stage; P, P-value; SD, standard deviation. Stages 1 to 8 indicate the scores of Demirjian developmental stages A to H.

*$P < 0.01$ and **$P < 0.001$ indicate a statistically significant difference between Korean and Japanese population data

the Japanese population revealed adjusted $r^2$ values of 0.682 and 0.585, which is similar to the results for all four M2s and M3s. However, the values of adjusted $r^2$ decreased when combining M2s and M3s (0.517–0.662), and the lowest value was observed when combining maxillary and mandibular M2s (0.448 in males, 0.296 in females) in the Japanese population (S4 Table).

## Discussion

The present study aimed at validating Lee's age estimation method for confirmation of applicability when performing forensic practices and assessing the 18-year threshold based on the estimated age by Lee's method in the Korean and Japanese populations. The validity of Lee's method was tested by calculating the correlation coefficients between estimated and

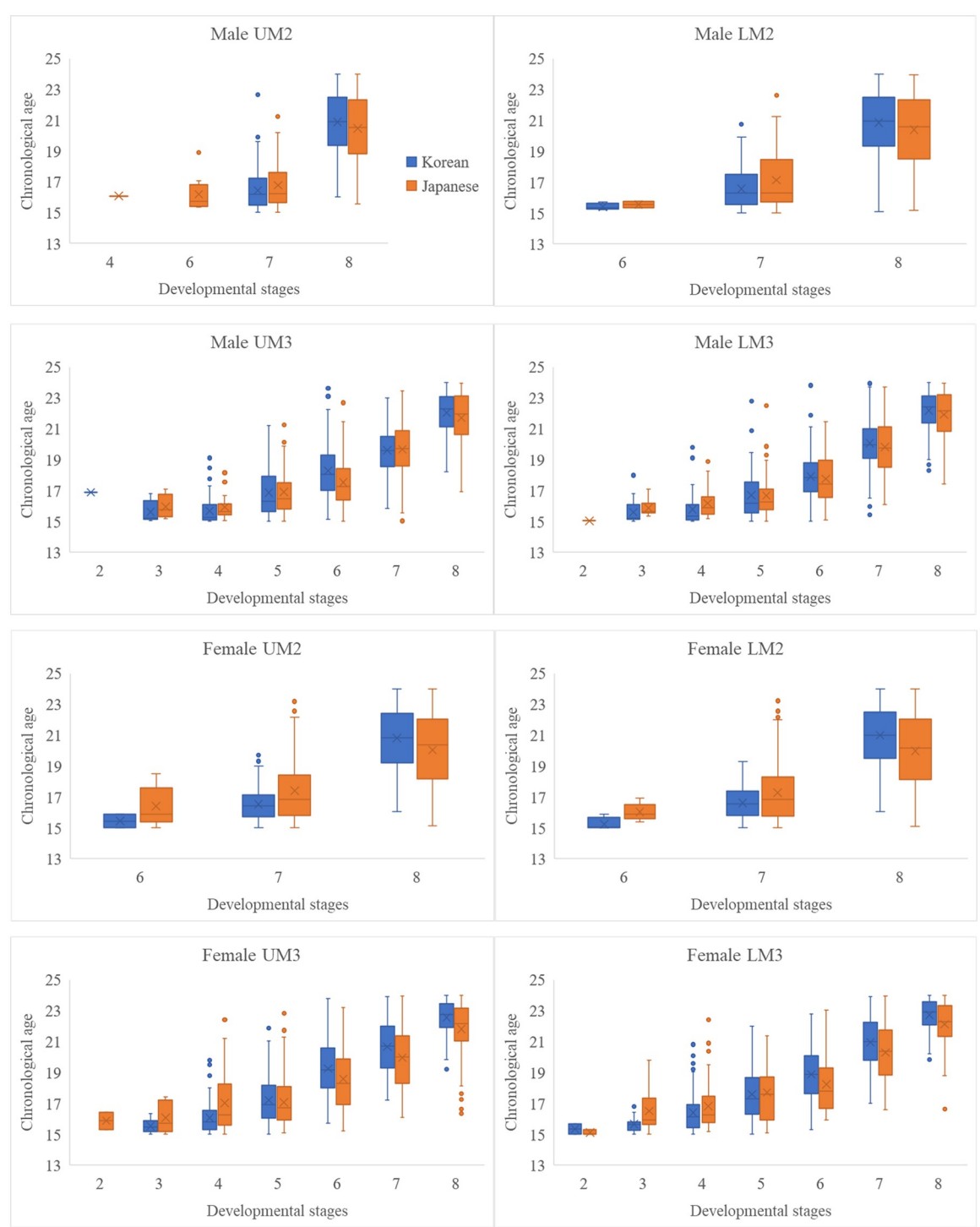

**Fig 1. Box plots of the relationship between chronological age and developmental stage for M2s and M3s in both jaws.** U, maxilla; L, mandible; Stages 1 to 8 indicate the scores of Demirjian developmental stages A to H. Blue and orange represent the Korean and Japanese populations, respectively.

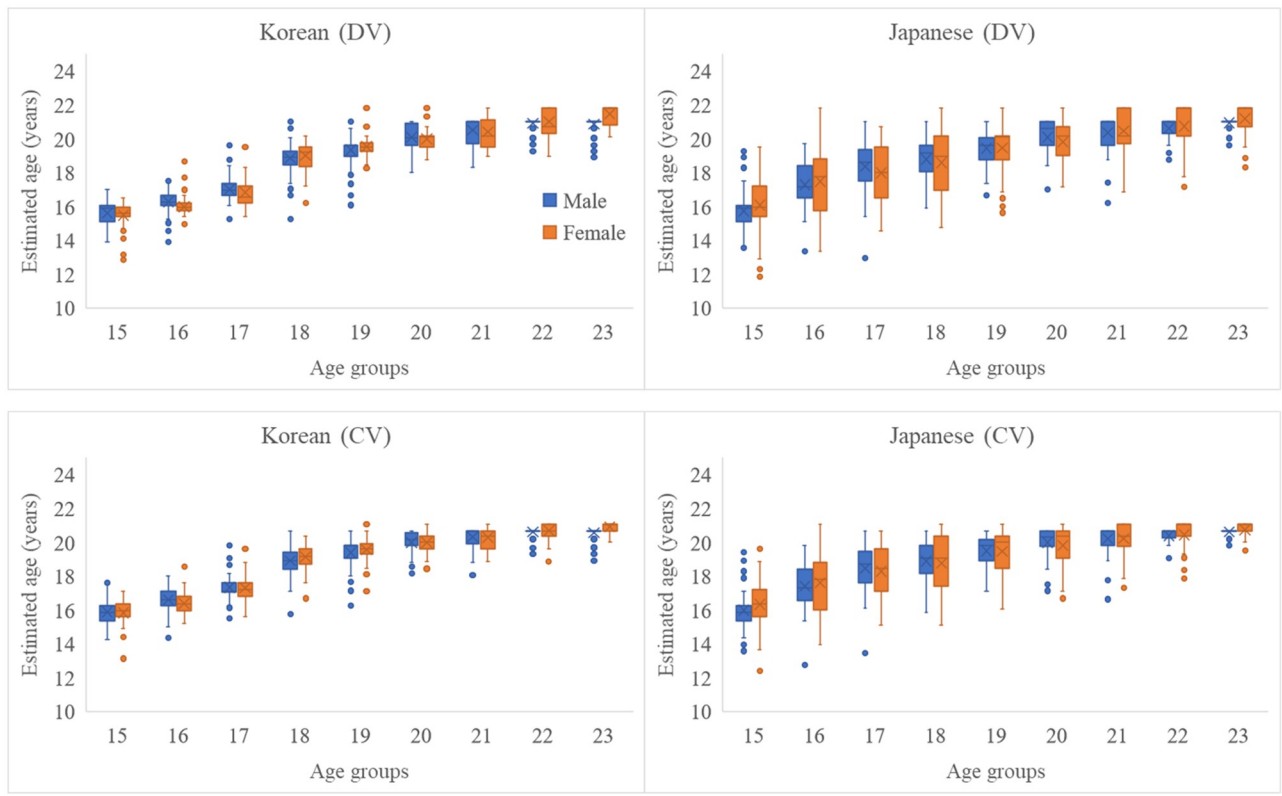

**Fig 2. Box plots for comparison between chronological and estimated ages.** The age was estimated based on the maturity of M2s and M3s in both jaws, and analyses were performed based on the assumption that the variables were discrete (top) and continuous (bottom). Blue represents male, and orange represents female.

chronologic ages, and the sensitivity and specificity were calculated by setting a threshold based on the age of 18 years for both populations. Further, we compared the differences in the chronological age according to the tooth development stage between the Korean and Japanese populations by regression analysis.

Demirjian's scheme is a widely used standard in M3 research for evaluating the developmental stage of teeth due to its objectivity in only evaluating anatomical features [6]. Olze et al. [26] evaluated the mineralization of tooth 38 using the methods of Gleiser and Hunt, Demirjian et al., Gustafson and Koch, Harris and Nortje, and Kullman et al. and reported that

**Table 4. Correlation coefficients between estimated and chronological ages.**

|  |  | Korean | | | Japanese | | |
|---|---|---|---|---|---|---|---|
|  |  | **Male** | **Female** | **Total** | **Male** | **Female** | **Total** |
| DV | PCC (95% CI) | 0.90 (0.89–0.92) | 0.92 (0.91–0.93) | 0.91 (0.90–0.92) | 0.80 (0.76–0.83) | 0.72 (0.68–0.76) | 0.76 (0.73–0.78) |
|  | ICC (95% CI) | 0.92 (0.86–0.95) | 0.93 (0.87–0.96) | 0.93 (0.86–0.95) | 0.86 (0.83–0.89) | 0.83 (0.80–0.86) | 0.85 (0.82–0.87) |
| CV | PCC (95% CI) | 0.90 (0.88–0.91) | 0.91 (0.90–0.92) | 0.90 (0.89–0.91) | 0.79 (0.76–0.83) | 0.72 (0.67–0.76) | 0.75 (0.72–0.78) |
|  | ICC (95% CI) | 0.89 (0.84–0.93) | 0.91 (0.86–0.94) | 0.90 (0.85–0.93) | 0.84 (0.80–0.87) | 0.81 (0.78–0.84) | 0.83 (0.80–0.85) |

DV, discrete variable; CV, continuous variable; CI, confidence interval. Pearson correlation coefficients (PCCs) and interclass correlation coefficients (ICCs) were calculated based on age estimation using M2s and M3s in both jaws. The data using single molar, or two combined molars are presented in S2 Table. All point estimates are $P < 0.001$.

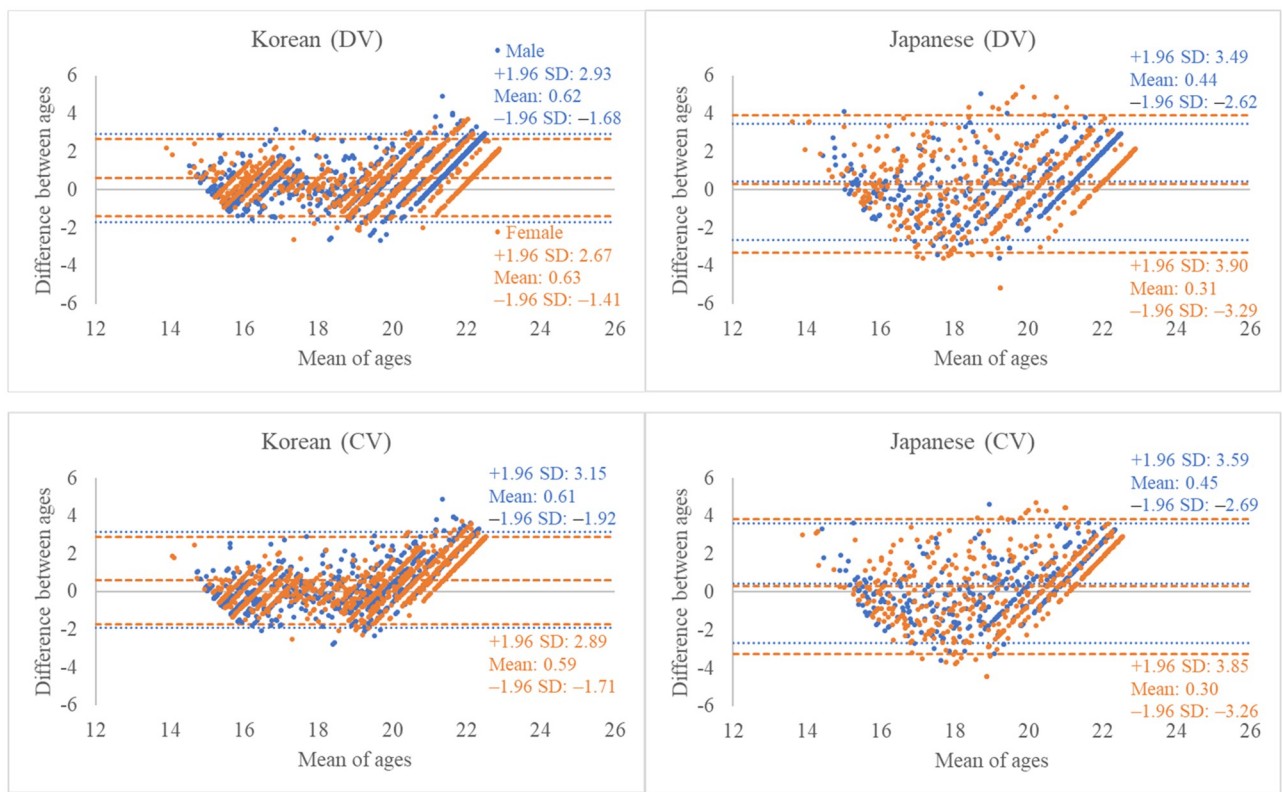

**Fig 3. Bland–Altman plot of the difference between chronological and estimated ages.** The age was estimated based on the maturity of M2s and M3s in both jaws. The difference was calculated by subtracting the estimated age from chronological age. DV, discrete variable; CV, continuous variable. Blue represents male, and orange represents female.

Demirjian's classification system resulted in the highest accuracy. Dhanjal et al. [27] also reported that Demirjian's classification system had better reproducibility compared to that of Moorrees and Haavikko. This study therefore employed Demirjian's system to evaluate the maturity of M2s and M3s, with almost perfect intra- and interobserver reliability.

**Table 5. Classification performance based on the 18-year threshold.**

| | | Korean | | | Japanese | | |
|---|---|---|---|---|---|---|---|
| | | **Male (%)** | **Female (%)** | **Total (%)** | **Male (%)** | **Female (%)** | **Total (%)** |
| DV | Sen (95% CI) | 97.3 (95.7–98.5) | 97.0 (95.3–98.2) | 97.2 (96.1–98.0) | 93.0 (89.3–95.7) | 85.3 (80.5–89.2) | 89.1 (86.2–91.6) |
| | Spe (95% CI) | 95.3 (92.3–97.4) | 98.0 (95.7–99.3) | 96.7 (94.9–98.0) | 67.2 (58.5–75.0) | 68.2 (60.7–75.1) | 67.8 (62.2–73.0) |
| | PPV (95% CI) | 97.7 (96.1–98.7) | 99.0 (97.8–99.6) | 98.3 (97.4–99.0) | 85.2 (80.6–89.0) | 81.2 (76.2–85.5) | 83.2 (79.9–86.1) |
| | NPV (95% CI) | 94.7 (91.5–96.9) | 94.2 (91.0–96.5) | 94.5 (92.3–96.1) | 82.6 (74.1–89.2) | 74.2 (66.7–80.8) | 77.6 (72.1–82.5) |
| | Acc (95% CI) | 96.7 (95.3–97.7) | 97.3 (96.1–98.3) | 97.0 (96.1–97.7) | 84.5 (80.6–87.9) | 78.7 (74.6–82.4) | 81.4 (78.7–84.0) |
| CV | Sen (95% CI) | 97.8 (96.3–98.8) | 98.7 (97.4–99.4) | 98.3 (97.3–98.9) | 95.2 (92.0–97.4) | 87.1 (82.5–90.8) | 91.1 (88.4–93.3) |
| | Spe (95% CI) | 94.7 (91.5–96.9) | 94.7 (91.5–96.9) | 94.7 (92.6–96.3) | 67.2 (58.5–75.0) | 64.2 (56.5–71.3) | 65.5 (59.9–70.8) |
| | PPV (95% CI) | 97.3 (95.7–98.5) | 97.4 (95.8–98.5) | 97.4 (96.3–98.2) | 85.5 (81.0–89.2) | 79.6 (74.6–84.0) | 82.5 (79.3–85.5) |
| | NPV (95% CI) | 95.6 (92.6–97.6) | 97.3 (94.7–98.8) | 96.4 (94.6–97.8) | 87.4 (79.4–93.1) | 75.5 (67.7–82.2) | 80.4 (74.9–85.1) |
| | Acc (95% CI) | 96.8 (95.4–97.8) | 97.3 (96.1–98.3) | 97.1 (96.2–97.8) | 86.0 (82.2–89.2) | 78.3 (74.2–82.0) | 81.9 (79.2–84.4) |

DV, discrete variable; CV, continuous variable; CI, confidence interval; Sen, sensitivity; Spe, specificity; PPV, positive predictive value; NPV, negative predictive value; Acc, accuracy. All point estimates are $P < 0.001$.

**Table 6. Intercepts and coefficients of regression regarding the maturity stages of M2s and M3s as discrete and continuous variables for Korean (K) and Japanese (J) data.**

(a)

| DV | Stg | Male (K) | Female (K) | Male (J) | Female (J) |
|---|---|---|---|---|---|
| Intercept | | 22.31 | 22.87 | 21.96 | 22.23 |
| UM2 | D (4) | | | −0.86* | |
| | F (6) | | −1.17* | −0.57* | −0.16* |
| | G (7) | −1.44 | −1.21 | −0.72 | −0.51 |
| | H (8) | 0 | 0 | 0 | 0 |
| UM3 | B (2) | −0.57* | | | −2.22* |
| | C (3) | −2.02 | −2.40 | −1.97 | −2.40 |
| | D (4) | −2.08 | −2.30 | −1.96 | −1.27 |
| | E (5) | −1.74 | −1.96 | −1.47 | −1.57 |
| | F (6) | −1.30 | −1.53 | −1.77 | −0.81 |
| | G (7) | −1.18 | −1.02 | −0.64 | −0.71 |
| | H (8) | 0 | 0 | 0 | |
| LM2 | F (6) | −0.98* | −2.24 | −0.73* | −1.10* |
| | G (7) | −0.95 | −1.71 | −0.44 | −0.42* |
| | H (8) | 0 | 0 | 0 | 0 |
| LM3 | B (2) | | −2.16 | −5.11 | −4.59 |
| | C (3) | −2.34 | −2.06 | −3.01 | −3.48 |
| | D (4) | −2.46 | −1.88 | −3.11 | −3.49 |
| | E (5) | −2.09 | −1.64 | −2.99 | −2.74 |
| | F (6) | −1.72 | −1.49 | −2.23 | −2.85 |
| | G (7) | −1.16 | −0.87 | −1.32 | −1.19 |
| | H (8) | 0 | 0 | 0 | 0 |
| adj. $r^2$ | | 0.834 | 0.855 | 0.693 | 0.596 |

(b)

| CV | Male (K) | Female (K) | Male (J) | Female (J) |
|---|---|---|---|---|
| Intercept | −5.78 | −7.28 | 3.41 | 5.70 |
| UM2 | 1.30 | 1.08 | 0.28* | 0.46 |
| UM3 | 0.56 | 0.69 | 0.63 | 0.40 |
| LM2 | 0.99 | 1.53 | 0.55 | 0.38* |
| LM3 | 0.61 | 0.41 | 0.82 | 0.76 |
| adj. $r^2$ | 0.804 | 0.834 | 0.656 | 0.563 |

(a) Multiple linear regression using discrete variables. The estimated age is calculated by adding an intercept and the numerical values equivalent in the stage of each tooth in each column. DV, discrete variable; CV, continuous variable; U, maxilla; L, mandible; Stg, stage; Stages 1 to 8 indicate the scores of Demirjian developmental stages A to H.

*$P > 0.05$, not statistically significant.

(b) Multiple linear regression using continuous variables. The estimated age is calculated by adding an intercept to the multiply of the stage of each tooth (1–8) and the numerical value on each column. DV, discrete variable; CV, continuous variable; U, maxilla; L, mandible;

*$P > 0.05$, not statistically significant.

Based on the strong correlations (0.90 or more in PCCs) between estimated and chrono-logic ages in the Korean population, we confirmed the appropriateness of Lee et al.'s formula for estimating the age of the Korean population. Meanwhile, when Lee's method was applied to the Japanese population, broader interquartile ranges of the estimated age (Fig 2, S1 Table), lower values of PCCs and ICCs (Table 4), and more scattering patterns in the Bland–

Altman plot were observed (Fig 3). These results suggest that the accuracy of estimation may decrease if age estimation methods based on the Korean population are used for the Japanese population and imply possible population differences in dental development between the Korean and Japanese populations. Extensive evidence suggests that age estimation should be performed based on population data to improve the accuracy of estimation [28–39]. Olze et al. [28] compared the developmental stages of M3 between German and Japanese individuals and reported significant differences in stages D, E, and F. Olze et al. [29] also assessed M3 mineralization in a Black African population and reported that Black Africans tended to reach mineralization stages earlier than Germans. Based on three different population specific reference datasets, Jayaraman et al. [30] tested the accuracy of age estimation in southern Chinese, and reported highest accuracy when estimation was performed based on the same population. The results of these studies indicate a difference in tooth development among different populations. However, several studies have also been reported with the opposite results. Liversidge et al. [14, 40] examined whether ethnic differences existed in tooth maturity and reported that the accuracy of estimation was not vivid regardless of the specific reference from different population data. They asserted that the error of estimated ages which was reported by many population-specific researches, may be from the different sample distributions, and not from population differences. Rodriguez et al. [41] tested the validity of twelve age estimation methods which was based from non-mexican population data or international multi-population data in Mexican children. They reported quite good applicability of the tested methods and argued that the population differences in teeth development may be very small. Although the values of coefficients between the estimated and chronological ages in the Japanese population was lower in this study, it could also be evaluated as a strong correlation (0.80 to 0.72). In addition, we should consider that this study was analyzed with radiographs collected from a single institution from both countries. Therefore, generalization of possible population differences between the two populations based on the own results of this study to be looked at very carefully. If we want to confirm the results from this study and generalize the tendency to the entire population, future studies with data from multiple institutions should be performed. Further, Japanese-based studies are warranted for accurate age estimation of the Japanese population.

Analysis of the performance of the 18-year threshold classification revealed that the sensitivity was similar for both populations, but specificity was lower for the Japanese population (65.5–67.8%) than the Koreans (94.7–96.7%), indicating a lower probability of correctly judging a minor as minor, which may lead to errors in the legal process. We conjecture that the reason for the lower specificity in the Japanese population may be the tendency for higher variability in maxillary and mandibular M2s. The coefficients of determination (adjusted $r^2$) between the stages of M2s and M3s and chronological age were lower in the Japanese population (0.596–0.693) than in the Korean population (0.834–0.855). Based on the results of regression analysis for the Japanese population, the coefficients of determination were decreased if the regression was performed with the combined developmental data of M3s and M2s, and the least values of adjusted $r^2$ were observed on the regression with the combined developmental data of upper and lower M2s. However, given the lack of comparable studies on the developmental stages of M2s in the Japanese population, it is difficult to conclusively determine whether the Japanese population exhibits unique M2 developmental patterns. Population studies observing the developmental chronology of M2s with larger samples in the Japanese population are warranted to confirm these findings.

Previous studies have analyzed the chronology of M3s according to M3 developmental stages in the Japanese population. Olze et al. [28] examined mineralization stages in relation to age in a Japanese population. In their study, the mean ages of E, F and G stages of M3 of both

jaws were older than this study. In Olze's study, a large number of individuals at a relatively old age were identified in the E, F and G stages, indicative of higher values than the mean age and standard deviation observed in this study. Arany et al. [42] also surveyed the development of M3s in Japanese juveniles and reported the mean values of each stage of M3s that were similar to the results of this study, with the exception of stage G of maxillary and mandibular M3s in males and stage G of maxillary M3 in females. Nevertheless, the sample size and scope of the studies were different, which precludes direct comparisons of the two studies. In this regard, there is a need to analyze the developmental chronology of M3s using larger Japanese samples to clarify the developmental patterns of M3s.

Several studies have analyzed the developmental chronology of M3 in the Korean population. In a study by Lee et al. [4], the mean age and standard deviation in the E, F, and G stages of M2s and M3s were similar to those in this study. The mean age observed in this study was higher than the mean age of M2s and M3s reported in another Korean-based study for almost all stages of M3s [43]. These discrepancies could be due to the different age ranges assessed in the studies. In this regard, the age range in this study was between 15 and 23 years, which is narrower than that in previous studies (4–26 years).

Lee et al. [4] performed regression analysis with discrete and continuous data, but no significant difference was observed. As such, they were unable to conclude which data type was more suitable for classifying the stages of teeth. Similarly, in this study, only a small difference was noted between discrete and continuous data in PCCs and ICCs, ranging from 0.01% to 0.03%. Moreover, only small differences were observed in classification performance (0.1–2.8%) and results of regression analysis coefficients for discrete and continuous data (0.021–0.037). The $r^2$ value was slightly lower than that in a previous study. This could be because the age span of the previous study was 3–23 years, which is wider than the age span of the present study (15–23 years). The previous study also included all developmental stages of M2s and M3s from stages 1 to 8. In contrast, the present study included fewer stages, which may have underpinned the lower coefficients of determination.

To determine whether a living individual is an adult, the European Asylum Support Office and Study Group on Forensic Age Diagnostic recommend methods that include wrist, clavicle, and dental examinations, including wisdom teeth [44, 45]. For the Korean forensic practice, the population studies about clavicles [46] and third molars [4, 43, 47] were already published and used in many age estimation cases. However, a population study of wrist examination in the Koreans has not been conducted to date. To fulfill the international standards for living Korean individuals and establish an estimation protocol, future studies should evaluate the development of hand wrist bones in the Korean population.

## Conclusions

Our study demonstrated that Lee's method might be suitable for age estimation in the Korean population. However, the accuracy and specificity for Japanese juveniles seemed to be insufficient for application in forensic practice. Considering the limitations of this study which was conducted using data from a single institution, follow-up studies using data collected from multiple institutions are needed to be able to generalize the results from this study. Population studies using larger Japanese samples should be conducted to develop Japanese-specific age estimation approaches. Our findings suggested that the distinctness of tooth development, especially M2, in the Japanese population. Future studies on the chronology of permanent tooth development are warranted to clarify the developmental patterns of teeth in Japanese children.

## Supporting information

**S1 Table. Median and interquartile range of estimated ages according to chronologic age group.**
(PDF)

**S2 Table. Correlation coefficient between estimated and chronological age.**
(PDF)

**S3 Table. Intercepts and coefficients of regression regarding the maturity stages of M2s and M3s as discrete and continuous variables for Korean data.**
(PDF)

**S4 Table. Intercepts and coefficients of regression regarding the maturity stages of M2s and M3s as discrete and continuous variables for Japanese data.**
(PDF)

**S1 Fig. Box plots for comparison between chronological and estimated ages.**
(PDF)

## Acknowledgments

The authors wish to acknowledge statistical consultation supported by the Department of Biostatistics of the Catholic Research Coordinating Center.

## Author Contributions

**Conceptualization:** Sang-Seob Lee.

**Data curation:** Akiko Kumagai, Sin-Young Kim, Sang-Seob Lee.

**Formal analysis:** Sang-Seob Lee.

**Funding acquisition:** Sang-Seob Lee.

**Investigation:** Sehyun Oh, Akiko Kumagai, Sang-Seob Lee.

**Methodology:** Sang-Seob Lee.

**Writing – original draft:** Sehyun Oh.

**Writing – review & editing:** Sehyun Oh, Sang-Seob Lee.

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
