## [Decision Letter · Decision Letter 0]

1 Apr 2022

PONE-D-22-06902Accuracy of dental age estimation and assessment of the 18-year threshold based on the development of second and third molars in Korean and Japanese populationsPLOS ONE

Dear Dr. Lee,

Thank you for submitting your manuscript to PLOS ONE. After careful consideration, we feel that it has merit but does not fully meet PLOS ONE’s publication criteria as it currently stands. Therefore, we invite you to submit a revised version of the manuscript that addresses the points raised during the review process.

We look forward to receiving your revised manuscript.

Kind regards,

Dinh-Toi Chu, PhD

Academic Editor

PLOS ONE

----------------

Reviewer #1: Dear editor, thank you for the opportunity you gave me to assess the scientific quality of a paper titled “Accuracy of dental age estimation and assessment of the 18-year threshold based on the development of second and third molars in Korean and Japanese populations”. My comments and suggestions are presented below.

Dear authors, thank you for addressing an important topic that combine anatomical and clinical data. Please try to look at your methods and conclusion parts seriously.

Introduction

In the introduction section the statement “In addition, to improve practicality, age estimation using one, two, and four teeth was presented,…” is vague. Please clearly define what teeth one, two and four mean in terms of scientific name.

Methods

Why there is no clear sample size calculation and sampling methods?

The samples were collected from only one institution in both countries. How the authors can conclude based on this institutional data to the general Korean or Japanese population? It is very difficult to generalize. The authors should reconsider their generalization.

In line 95 “The number of subjects assigned to each stage was distributed equally to reduce the bias error in the statistical analysis” this doesn’t work for Japanese data. Please look at it.

Who reads the radiographs? The authors should state the experience and profession.

Discussion

The last paragraph is not relevant to your topic.

Conclusion

Your conclusion, Lee’s method was suitable for age estimation in the Korean population or for Japanese ……. You lack data to reach on this conclusion. Look the methods section.

Reviewer #2: This was an interesting article which asses the 18-year threshold, compare any potential different between the Japanese and Korean population in estimation of age from maxillary and mandibular second and third molars. It evaluated Lee’s age estimation method for the Korean population also. It concludes that Lee’s method was appropriate for age estimation in the Korean population. But for Japanese it was less accurate for application in forensic practice. The analysis of the performance of the 18-year threshold classification revealed that high sensitivity and specificity for Koreans.

This minor issue has to be clarified by the author.

• Describing sample size estimation

• How endocrine disorders are excluded in the study?

• State about the assumption of statically analysis methods and how it was assessed

• Better if the result is presented with sub headings. It needs to be clear for readers.

• State descriptive statistics before going to regressions

• The paragraph from line 234-238 better suits to the introduction not in the discussion

• The interquartile range was not described in the result section but discussed in the discussion part (line number 244-45)

Reviewer #3: Journal: PLOS ONE

Title: Accuracy of dental age estimation and assessment of the 18-year threshold based on the development of second and third molars in Korean and Japanese populations

Manuscript Number: PONE-D-22-06902

The manuscript describes ‘accuracy of dental age estimation and assessment of the 18-year threshold based on the development of second and third molars in Korean and Japanese populations. The research design is very well defined and accurate for intended objectives. The results are very interesting and add valuable knowledge to the field. However, some revision needs to be done prior to its publication.

1. The title is too log please make it SMART.

2. In the introduction section, from line 77-80, it says however, (no study to date has conducted a direct comparison with Korean population data in this regard.

This study aimed to validate Lee’s method for age estimation in Korean and Japanese populations.

It's not clear. The gap is that no research is being conducted on the topic, and the gap you intend to fill is to 'validate Lee's method for age estimation.' Make it perfectly clear!!!

Make it obvious how your research differs from this one. (https://doi.org/10.1016/j.forsciint.2010.04.054).

3. Page 6 line 122 ‘Intra- and interobserver reliabilities were calculated using Cohen’s kappa statistics. It also needs to be clear to readers and include citations.

4. Page 6, line 124, explain why you chose 'Pearson correlation coefficients (PCCs)' and why Spearman's rank correlation coefficient was not used.

5. You stated on page 7, line 138, that you used' multiple linear regression.' Have you checked the linear regression estimator assumptions before using it?

6. The result, discussion and recommendations were well written.

Reviewer #4: The article is interesting but some major correction needs to be carried out to add to the quality of the manuscript. Find below some of my observation regarding the manuscript.

Abstract

Line 18 and 19, the aim of the study is not clear. :” investigate the potential differences between the Korean and Japanese populations” in what? Also your objective should be in line with the title of the manuscript. The title and /or aim need to be modified to agree with each other. What is the rational of validating Lees’s methods?

Line 21 “15–23 years …” add mean and standard deviation of the age”

Line 25 the word correlation should be changed to relationship

Line 26 “Our analyses” changed to Our results

Lines 26-32, you need to enrich and robust your results sections. For example it is not enough to report that there is weak correlation. Scientific community will be interested to know if the weak correlation is significant or not. Also, remember correlation is influenced by the sample size, with higher sample size weak correlation can reveal significant correlation; similarly a moderate to even strong correlation can reveal insignificant correlation with smaller sample size. The used of Lee’s method should captured in the methods section. There is no section to signify the application of multiple regressions in your results section.

Line 32 “Our results suggest” changed to In conclusion,

Line 33 “age estimation method” be specific, which of age estimation method.

General comment: Please streamline your title, aim and conclusion to in be in agreement

“Keywords” are completely missing in the manuscript.

Introduction

Lines 58 and 59 “To increase objectivity, the developmental …” please the sentence needs citation.

Line 79; are validating or employing Lee’s methods?

Materials and Methods

Line 86, refer to my comment in abstract section on age.

Lines 93 and 94 “of the samples” please change to study population. Also, “Chronological age was calculated as ...” any reference to back this method of chronological age estimation?

Line 102 “birth date” date of birth

Line 120 “3 months” three months

General comment: the statistical analyses section should be in standalone paragraph. Some fundamental criteria in chosen and reporting statistical analyses were ignored. For example, you test your data for normality distribution so as to be guide on which of the statistical test (parametric or non parametric) to be applied and also, to know which of the measure of central tendency (mean or median) or dispersion (standard deviation, inter quartile etc) to be used in summarizing your results. The acceptable level of significance and confidence level should be stated. The order and chronology of the statistical test should be from simple to complex. The multiple regressions should used to depict relationships.

Results

Lines 143 and 148 “The results of the statistical analysis…” there is better way to present results. You should report the important components of the tables/figures the make reference to them.

Line 153, nothing like “excellent correlation” it is moderate or strong. Also, emphasize on the significant please.

Line 161, box plot for comparison or differences between group not really relationship plot.

Lines 161-164, “The age was …female” is this part of the legend”? if yes please bold it. The same applies to lines 166 to 169.

General comment:

You need to reorganize your results from simple to complex. For example you can start with descriptive statistics, comparison results, then the correlation and relationship results will be at the tail end. The results should also follow the order of the objectives of the study.

Please minimize the unnecessary horizontal line in your tables.

Tables of correlations and specificity, … lack associated P values.

Tables 5 and 6: the titles are inappropriate. Let the title depict the content of the tables. Avoid the words “statistical data”. The whole of the table need rearrangement, consider landscaping your tables.

Discussion

Line 238, why revalidating?

Lines 243 to 256 are part of results not discussion.

General comment:

Most of the stuff here are results. The authors should lay more emphases on the implication of their findings. The scientific bases of the observed results need to also be discussed. Limitation and recommendation (based on the applicability of the study) can be stated at the tail end of this section.

NOTE: If reviewer comments were submitted as an attachment file, they will be attached to this email and accessible via the submission site. Please log into your account, locate the manuscript record, and check for the action link "View Attachments". If this link does not appear, there are no attachment files.]

---

## [Author Response · Author response to Decision Letter 0]

16 Apr 2022

April 16, 2022

Dr. Dinh-Toi Chu

Academic Editor

PLOS ONE

Dear Dr. Chu:

I appreciate the reviewers of ‘PLOS ONE’ for taking their time to review my manuscript (Manuscript-ID; PONE-D-22-06902) entitled "Accuracy of dental age estimation and assessment of the 18-year threshold based on the development of second and third molars in Korean and Japanese populations". I have made some corrections and clarifications in the manuscript after going through the reviewers’ comments. The revised manuscript is included with the revision been indicated by yellow highlight. The changes and the point-by-point responses are summarized below.

I hope that you will now find our paper suitable for publication in your esteemed journal. However, if any further changes are needed, we shall be happy to consider.

Thank you for your consideration. I look forward to hearing from you.

Sincerely yours,

Sang-Seob Lee, D.D.S., Ph.D.

 Department of Anatomy, College of Medicine, The Catholic University of Korea, 

 222, Banpo-daero, Seocho-gu, Seoul, 06591, Republic of Korea

 Tel: +82-2-2258-7672

 Fax: +82-2-537-7081

E-mail; sslee1418@gmail.com

Reviewer: 1 

Comments to the Author 

Thank you for addressing an important topic that combine anatomical and clinical data. Please try to look at your methods and conclusion parts seriously.

#1. In the introduction section the statement “In addition, to improve practicality, age estimation using one, two, and four teeth was presented,…” is vague. Please clearly define what teeth one, two and four mean in terms of scientific name. 

- Response: We totally agree with and appreciate the reviewer’s opinion. In response to the reviewer’s comments, we have modified the manuscript for clear definition of the names of teeth. Thank you for your comment.

- Corrections: They suggested various regression models using the developmental stage of a single tooth (M2 or M3), using stages of two teeth among M2s and M3s, and using stages of all M2s and M3s in both jaws, as variables. In addition, they treated variables as discrete or continuous, and the results of both types of variables were compared. (Line 67~70)

#2. Why there is no clear sample size calculation and sampling methods? 

- Response: We totally agree with and appreciate the reviewer’s opinion. In response to the reviewer’s comments, we have inserted sentences in the materials and methods section of the manuscript which indicate the methods and results of the sample size estimation. Thank you for your comment.

- Corrections: The sample size was estimated based on multiple linear regression considering four predictors, with a confidence level of 95%, and a statistical power of 80. Supposing the effect size f2 was small (i.e., f2 = 0.03), the minimum sample size was calculated as 403 using the G*Power software (Dusseldorf, Germany). (Line 98~101)

#3. The samples were collected from only one institution in both countries. How the authors can conclude based on this institutional data to the general Korean or Japanese population? It is very difficult to generalize. The authors should reconsider their generalization. 

- Response: We totally agree with and appreciate the reviewer’s opinion. That is a clear limitation of our study. We need to be careful in interpreting and generalizing our findings to the population. In response to the reviewer’s comments, we have inserted some sentences which indicate the limitation of this study, and the need of future research with data from multiple institutions, in the discussion section of the manuscript. Also, we have modified some expressions in the conclusion section. Thank you for your valuable opinion.

- Corrections: In addition, we should consider that this study was analyzed with radiographs collected from a single institution from both countries. Therefore, generalization of possible population differences between the two populations based on the own results of this study to be looked at very carefully. If we want to confirm the results from this study and generalize the tendency to the entire population, future studies with data from multiple institutions should be performed. (Line 336~341)

Our study demonstrated that Lee’s method might be suitable for age estimation in the Korean population. However, the accuracy and specificity for Japanese juveniles seemed to be insufficient for application in forensic practice. Considering the limitations of this study which was conducted using data from a single institution, follow-up studies using data collected from multiple institutions are needed to be able to generalize the results from this study. (Line 400~404)

#4. In line 95 “The number of subjects assigned to each stage was distributed equally to reduce the bias error in the statistical analysis” this doesn’t work for Japanese data. Please look at it.

- Response: We totally agree with and appreciate the reviewer’s opinion. In response to the reviewer’s comments, we have revised the sentence to correct the expression.

- Corrections: The number of subjects assigned to each stage was distributed similarly (around one hundred for Koreans and fifty for Japanese) to reduce the bias error in the statistical analysis. (Line 114~116)

#5. Who reads the radiographs? The authors should state the experience and profession. 

- Response: We totally agree with and appreciate the reviewer’s opinion. In response to the reviewer’s comments, we have added a sentence stating the observers’ experience and profession. Thank you for your valuable opinion.

- Corrections: Both observers were well-trained forensic odontologists with experience in forensic examination for more than ten years. (Line 133~134)

#6. The last paragraph (of discussion section) is not relevant to your topic. 

- Response: Thank you for your suggestion. This study was conducted to improve the accuracy of age estimation of subjects in their late teens to young adults. This paragraph has been inserted to suggest the direction of future research projects to improve accuracy of estimation in the forensic practices. In response to the reviewer’s comments, the sentences in the paragraph have been made more concise to emphasize the need for further research. Thanks to your comments; what we intend to suggest in this paragraph have become clearer as a result of the modifications made. Thank you very much.

- Corrections: For the Korean forensic practice, the population studies about clavicles [46] and third molars [4,43,47] were already published and used in many age estimation cases. However, a population study of wrist examination in the Koreans has not been conducted to date. (Line 391~394)

#7. Your conclusion, Lee’s method was suitable for age estimation in the Korean population or for Japanese ……. You lack data to reach on this conclusion. Look the methods section

- Response: We totally agree with and appreciate the reviewer’s opinion. As we stated in the response for comment #3, that is a clear limitation of our study. In response to the reviewer’s comments, we have modified some expressions in the conclusion section. Thanks to your kind and dedicated review, the manuscript of this study is much more systematic and logical. We greatly appreciate your efforts.

- Corrections: Our study demonstrated that Lee’s method might be suitable for age estimation in the Korean population. However, the accuracy and specificity for Japanese juveniles seemed to be insufficient for application in forensic practice. Considering the limitations of this study which was conducted using data from a single institution, follow-up studies using data collected from multiple institutions are needed to be able to generalize the results from this study. (Line 400~404)

 

Reviewer: 2 

Comments to the Author

This was an interesting article which asses the 18-year threshold, compare any potential different between the Japanese and Korean population in estimation of age from maxillary and mandibular second and third

molars. It evaluated Lee’s age estimation method for the Korean population also. It concludes that Lee’s method was appropriate for age estimation in the Korean population. But for Japanese it was less accurate for application in forensic practice. The analysis of the performance of the 18-year threshold classification revealed that high sensitivity and specificity for Koreans. This minor issue has to be clarified by the author.

#8. Describing sample size estimation. 

- Response: We totally agree with and appreciate the reviewer’s opinion. In response to the reviewer’s comments, we have inserted sentences in the materials and methods section of the manuscript which indicate the methods and results of the sample size estimation. Thank you for your comment.

- Corrections: The sample size was estimated based on multiple linear regression considering four predictors, with a confidence level of 95%, and a statistical power of 80. Supposing the effect size f2 was small (i.e., f2 = 0.03), the minimum sample size was calculated as 403 using the G*Power software (Dusseldorf, Germany). (Line 98~101)

#9. How endocrine disorders are excluded in the study?

- Response: We appreciate the reviewer’s opinion. In response to the reviewer’s comments, the sentences were revised for clarifying the exclusion criteria, in the materials and methods section of manuscript. We are thankful for pointing out such an important point.

- Corrections: Radiographs showing severe tooth decay, endodontic treatment, pathological lesions in the jawbone, agenesia, and unclear images were excluded. Radiographs belonging to patients with a medical history of endocrinal disorders were also excluded. (Line 109~111)

#10. State about the assumption of statically analysis methods and how it was assessed. 

- Response: We totally agree with and appreciate the reviewer’s opinion. In response to the reviewer’s comments, we have added the sentences indicating the methods and results about the assumption of normality for the variables in the materials and methods section. Thank you for your valuable opinion.

- Corrections: Before statistical analysis, the assumption of normality was checked based on the values of skewness and kurtosis and the Shapiro-Wilk test. All continuous variables had univariate normality, as shown by skew values < 2 and kurtosis values < 7. (Line 147~150)

#11. Better if the result is presented with sub headings. It needs to be clear for readers.

- Response: We totally agree with your suggestion. In response to the reviewer’s suggestion, the ‘results’ section has been reorganized with sub-headings. In addition, we have also reorganized the ‘materials and methods’ section with sub-headings. Thank you for your valuable comments. Your suggestion makes the manuscript easier for our readers to understand.

- Corrections: Please refer to the revised manuscript.

#12. State descriptive statistics before going to regressions.

- Response: We totally agree with your suggestion. In response to the reviewer’s suggestion, the ‘results’ section has been reorganized in the order of ‘Datasets’, ‘Observer reliabilities’, ‘Descriptive statistics of M2s and M3s’, ‘Accuracy of estimated ages’, ‘Assessment of 18-year threshold’, and ‘Relationships between dental maturity and chronological ages’. Thank you for your valuable comments. Your suggestion makes the manuscript easier for our readers to understand.

- Corrections: Please refer to the revised manuscript.

#13. The paragraph from line 234-238 better suits to the introduction not in the discussion.

- Response: We totally agree with your suggestion. In response to the reviewer’s suggestion, this paragraph was moved to the ‘introduction’ section. Thank you for your valuable comment. The logical flow of the manuscript was enhanced with your suggestion.

- Corrections: Please refer to the revised manuscript.

#14. The interquartile range was not described in the result section but discussed in the discussion part (line number 244-45)

- Response: We totally agree with your point. The data including the interquartile range was shown in Figure 1 as a type of box plots. However, the statistical data which was used to make this figure had been omitted from the manuscript. In response to the reviewer’s suggestion, we made another table containing the statistical data of the estimated age according to the groups of chronological age and presented it as supporting information (S1 Table). We have also indicated the figure number in the text. Thanks to your kind and dedicated review, the manuscript of this study is much more systematic and logical. We greatly appreciate your efforts.

- Corrections: The interquartile ranges of estimated age were broader for the Japanese than that of the Koreans (Fig 2, S1 Table) and this tendency was observed more apparently in age groups under 18 years of age. (Line 222~224)

when Lee’s method was applied to the Japanese population, broader interquartile ranges of the estimated age (Fig 2, S1 Table), lower values of PCCs and ICCs (Table 4), and more scattering patterns in the Bland–Altman plot were observed (Fig 3). (Line 310~313)

 

Reviewer: 3 

Comments to the Author 

The manuscript describes ‘accuracy of dental age estimation and assessment of the 18-year threshold based on the development of second and third molars in Korean and Japanese populations. The research design is very well defined and accurate for intended objectives. The results are very interesting and add valuable knowledge to the field. However, some revision needs to be done prior to its publication 

#15. The title is too log please make it SMART. 

- Response: We totally agree with and appreciate the reviewer’s opinion. The title was shortened as much as possible. Thank you for your comment.

- Corrections: Accuracy of age estimation and assessment of the 18-year threshold based on second and third molar maturity in Koreans and Japanese (Line 1~3)

#16. In the introduction section, from line 77-80, it says however, (no study to date has conducted a direct comparison with Korean population data in this regard. This study aimed to validate Lee’s method for age estimation in Korean and Japanese populations. It's not clear. The gap is that no research is being conducted on the topic, and the gap you intend to fill is to 'validate Lee's method for age estimation.' Make it perfectly clear!!! Make it obvious how your research differs from this one. (https://doi.org/10.1016/j.forsciint.2010.04.054).

- Response: Thank you for your kind comments and for providing reference. We were not aware of the existence of the references you provided and were stating the wrong facts. Since the aim of this study is validation and assessment of the age threshold, we have revised some sentences in the ‘introduction’ section for emphasizing the need of a validation study for the Lee’s method for the Japanese population citing the article you introduced. Thanks to your valuable comments, a major error in this study could be eliminated and the logical flow could be enhanced. Thank you very much.

- Corrections: However, there was also a report presenting statistically significant differences in the development of the third molars between the Korean and the Japanese population [18]. Since these contrasting reports existed, it was necessary to verify whether the age estimation method based on the Korean population data is applicable to the Japanese population as well. (Line 85~89)

#17. Page 6 line 122 ‘Intra- and interobserver reliabilities were calculated using Cohen’s kappa statistics. It also needs to be clear to readers and include citations.

- Response: We totally agree with and appreciate the reviewer’s opinion. We have added a sentence which includes information about the observers’ experience and profession. The re-examination schedule by the second observer has also been described in the manuscript. The references about ‘Cohen’s kappa statistics’ have been cited.

- Corrections: Both observers were well-trained forensic odontologists with experience in forensic examination for more than ten years. One hundred randomly selected orthopantomograms were re-examined three months after the first examinations by the main and the second observer to assess intra- and interobserver reliabilities. (Line 133~136)

Intra- and interobserver reliabilities for the evaluation of maturity in M2s and M3s, were calculated using Cohen’s kappa statistics [23]. (Line 146~147)

The result of kappa for the intra-observer reliability was 0.949 (0.932 to 0.967, 95% confidence interval (CI)), while kappa for the inter-observer reliability was 0.942 (0.923 to 0.961, 95% CI). Both reliabilities were interpreted as “almost perfect” agreement according to the standard by Landis and Koch [24]. (Line 186~189)

Ref-23. Fleis JL, Cohen J. The equivalence of weighted kappa and the intraclass correlation coefficient as measures of reliability. Educ Psychol Meas. 1973;33: 613–619.

Ref-24. Landis JR, Koch GG. The measurement of observer agreement for categorical data. Biometrics. 1977;33: 159-174.

#18. Page 6, line 124, explain why you chose 'Pearson correlation coefficients (PCCs)' and why Spearman's rank

correlation coefficient was not used.

- Response: We totally agree with and appreciate the reviewer’s opinion. The estimated and chronological ages satisfied the normality assumption based on the values of skewness and kurtosis, and with the Shapiro-Wilk test. Therefore, we used Pearson correlation coefficient, which is a parametric test for calculating correlations between estimated and chronological ages. We have inserted a sentence indicating the methods and results about the assumption of normality for the variables in the materials and methods section. A brief expression has also been added explaining why we used Pearson correlation coefficients. Thank you so much. 

- Corrections: Before statistical analysis, the assumption of normality was checked based on the values of skewness and kurtosis and the Shapiro-Wilk test. All continuous variables had univariate normality, as shown by skew values < 2 and kurtosis values < 7. (Line 146~149)

Due to both estimated and chronological ages satisfied the normality assumption, we tested the Pearson correlation coefficients (PCCs) for observing correlations between the chronological and estimated ages. (Line 151~153)

#19. You stated on page 7, line 138, that you used' multiple linear regression.' Have you checked the linear regression estimator assumptions before using it?

- Response: Thank you for the question. Before the analysis, the chronological ages and the variables from developmental stages of M2s and M3s were checked and they satisfied the normality assumption. The quality of regression was also evaluated with a residual plot. We have added a sentence in the ‘materials and methods’ section about these explanations. Thanks to your kind and dedicated review, the manuscript of this study is much more systematic and logical. Thank you very much really.

- Corrections: Before statistical analysis, the assumption of normality was checked based on the values of skewness and kurtosis and the Shapiro-Wilk test. (Line 147~148)

Linear regression was performed to model the relationship between treating the developmental stage of each tooth and chronological age as dependent variables; a residual plot was used to test the quality of the model. (Line 165~167)

 

Reviewer: 4 

Comments to the Author 

The article is interesting but some major correction needs to be carried out to add to the quality of the

manuscript. Find below some of my observation regarding the manuscript.

#20. Line 18 and 19, the aim of the study is not clear. :” investigate the potential differences between the Korean and Japanese populations” in what? Also your objective should be in line with the title of the manuscript. The title and /or aim need to be modified to agree with each other. What is the rational of validating Lee’s methods? 

- Response: We totally agree with and appreciate the reviewer’s opinion. In response to the reviewer’s suggestion, we have set the purpose of the study as ‘validation’ and ‘assessment of year threshold’ in line with the study aims, title, and conclusion. We have modified the ‘title’, ‘abstract’ and conclusion accordingly. Thank you very much.

- Corrections: This study aimed to validate Lee’s age estimation method and assess the 18-year threshold in Korean and Japanese populations. (Line 18~19)

Accuracy of age estimation and assessment of the 18-year threshold based on second and third molar maturity in Koreans and Japanese (Line 1~3)

#21. Line 21 “15–23 years …” add mean and standard deviation of the age”

- Response: We totally agree with and appreciate the reviewer’s opinion. In response to the reviewer’s suggestion, we have added the mean and standard deviation of the age. Thank you very much.

- Corrections: We evaluated the maxillary and mandibular second (M2) and third molars (M3) in 2657 orthopantomograms of the Korean and Japanese populations aged 15–23 years (19.47±2.62 years for Koreans, 19.31±2.60 years for Japanese), using Demirjian’s criteria. (Line 19~22)

#22. Line 25 the word correlation should be changed to relationship

- Response: We totally agree with and appreciate the reviewer’s opinion. In response to the reviewer’s suggestion, we have revised the word to ‘relationship’. Thank you very much.

- Corrections: The relationship between developmental stage and chronologic age was analyzed using multiple linear regression. (Line 24~25)

#23. Line 26 “Our analyses” changed to Our results

- Response: We totally agree with and appreciate the reviewer’s opinion. In response to the reviewer’s suggestion, we have revised the words to ‘Our results’. Thank you very much.

- Corrections: Our results revealed that Lee’s method was appropriate for estimation in the Korean population. (Line 25~26)

#24. Lines 26-32, you need to enrich and robust your results sections. For example it is not enough to report that there is weak correlation. Scientific community will be interested to know if the weak correlation is significant or not. Also, remember correlation is influenced by the sample size, with higher sample size weak correlation can reveal significant correlation; similarly a moderate to even strong correlation can reveal insignificant correlation with smaller sample size. The used of Lee’s method should captured in the methods section. There is no section to signify the application of multiple regressions in your results section.

- Response: We totally agree with and appreciate the reviewer’s opinion. In response to the reviewer’s suggestions, we have revised the sentences in the abstract. We have also added sentences which indicate the methods and results about the assumption of normality for the variables in the materials and methods section. Thank you for your valuable opinion.

- Corrections: When the Lee’s method was applied to the Japanese population, a lower value of correlation coefficients between estimated and chronological age, and lower specificity were observed. (Line 26~28)

In the multiple linear regression between developmental stage and chronological age, lower values of adjusted r2 were observed in the Japanese population than in the Koreans. (Line 30~32)

Before statistical analysis, the assumption of normality was checked based on the values of skewness and kurtosis and the Shapiro-Wilk test. All continuous variables had univariate normality, as shown by skew values < 2 and kurtosis values < 7. (Line 147~150)

#25. Line 32 “Our results suggest” changed to In conclusion

- Response: We totally agree with and appreciate the reviewer’s opinion. In response to the reviewer’s suggestion, we have revised the words to ‘In conclusion,’. Thank you very much.

- Corrections: In conclusion, the Lee’s method derived from the Korean population data might be unsuitable for Japanese juveniles and adolescents. (Line 32~33)

#26. Line 33 “age estimation method” be specific, which of age estimation method.

- Response: We totally agree with and appreciate the reviewer’s opinion. In response to the reviewer’s suggestion, we have revised the sentences for specifying the hired method in this study. Thank you very much.

- Corrections: In conclusion, the Lee’s method derived from the Korean population data might be unsuitable for Japanese juveniles and adolescents. (Line 32~33)

#27. General comment (Abstract): Please streamline your title, aim and conclusion to in be in agreement.

- Response: We totally agree with and appreciate the reviewer’s opinion. In response to the reviewer’s suggestion, we have reconstructed or revised for streamlining the title, aim and conclusion not only in the ‘abstract’ section, but also in the ‘discussion’ section. Thank you for your valuable opinion.

- Corrections: Please refer to the revised manuscript.

#28. “Keywords” are completely missing in the manuscript.

- Response: Thank you for your suggestion. According to submission guidelines of Plos One, they do not ask for keywords in the manuscript. They just ask for keywords only to expedite the review process. In addition, these keywords are optional and not mandatory. We have checked over 100 published Plos One articles, and there was not a single article in which keywords were specified. I have attached the URL addresses for reference. Thank you very much.

https://journals.plos.org/plosone/s/submission-guidelines

https://journals.plos.org/plosone/s/submit-now

#29. Lines 58 and 59 “To increase objectivity, the developmental …” please the sentence needs citation.

- Response: We totally agree with and appreciate the reviewer’s opinion. In response to the reviewer’s comment, we have added citation at the end of the sentence. Thank you for comment.

- Corrections: To increase objectivity, the developmental stages of the second molar (M2) were combined to obtain the M3/M2 ratio, and multiple regression was performed to estimate age [11]. (Line 61~63)

#30. Line 79; are validating or employing Lee’s methods?

- Response: Thank you for the question. After your valuable comment, we reviewed the ‘introduction’ section and found that the need for validation and assessment of age threshold is not sufficiently explained in the manuscript. We have revised some sentences in the ‘introduction’ for clarifying our aims of this study. Your suggestions will make this manuscript more logical. Thank you very much.

- Corrections: The Lee’s method, after publication, was used in some cases for forensic age estimation in Korea, and some forensic practitioners demanded validation of the method using another Korean population data to prove the accuracy. Based on the Korean Civil Act and Juvenile Act, people aged over 18 years are considered adults [12,13]. Therefore, when the machinery of law, such as the courts, requested forensic examinations for discrimination of minor/adult status, they requested results based on the 18-year threshold. However, Lee et al. [4] did not determine adult probabilities based on the age of 18 years. For this reason, the study for assessment of the 18-year threshold based on the Korean population data became necessary. (Line 70~78)

Korea and Japan are neighboring countries, and genetic similarities between these two populations have been reported in genetic mapping studies of Asian populations [17]. However, there was also a report presenting statistically significant differences in the development of the third molars between the Korean and the Japanese population [18]. Since these contrasting reports existed, it was necessary to verify whether the age estimation method based on the Korean population data is applicable to the Japanese population as well. (Line 83~89)

#31. Line 86, refer to my comment in abstract section on age.

- Response: We totally agree with and appreciate the reviewer’s opinion. In response to the reviewer’s suggestion, we have added the mean and standard deviation of the age in the manuscript. We have also added a paragraph about datasets in the ‘results’ section for clearly revealing the statical data of the study samples. Thank you very much.

- Corrections: The final sample size was 2657 digital orthopantomograms which consisted of 900 males and 900 females from the Korean population, and 406 males and 451 females from the Japanese population (19.47±2.62 years for Koreans, 19.31±2.60 years for Japanese). (Line 101~104)

The mean chronological ages of the study populations were 19.47±2.62 years and 19.31±2.60 years for Koreans and Japanese (1800 Koreans and 857 Japanese), respectively. The mean chronological ages for males and females were 19.45±2.64 years and 19.49±2.59 years in the Korean population, and 19.48±2.60 years and 19.16±2.60 years in the Japanese population. The minimum and maximum ages of the Koreans included in the study populations were 15.00 years and 23.98 years in males, and 15.01 years and 23.97 years in females, respectively. For the Japanese, they were 15.01 years and 23.97 years in males, and 15.00 years and 23.99 years in females, respectively. No significant difference was observed in the chronological ages of females and males in the Korean and Japanese populations (P > 0.05). (Line 175~183)

#32. Lines 93 and 94 “of the samples” please change to study population. Also, “Chronological age was calculated as ...” any reference to back this method of chronological age estimation?

- Response: We totally agree with and appreciate the reviewer’s opinion. In response to the reviewer’s comments, we have revised the inadequate expression. The sentence which described the way of calculating chronological age was slightly modified and the references were added in which the chronological age was calculated with the same way as in this study. Thank you for the comment.

- Corrections: Table 1 shows the age and sex distribution of the study population. (Line 114)

The chronological age was calculated as the difference between the date of the radiography and the date of birth [19–21]. (Line 111~113)

#33. Line 102 “birth date” date of birth

- Response: We totally agree with and appreciate the reviewer’s correction. We have revised the faulty expression in the sentence. Thank you for your comment.

- Corrections: All collected data were anonymized, except sex, date of birth, and the date of taking radiographs. (Line 121~122)

#34. Line 120 “3 months” three months

- Response: We totally agree with and appreciate the reviewer’s correction. We have revised the faulty expression in the sentence. Thank you for comment.

- Corrections: One hundred randomly selected orthopantomograms were re-examined three months after the first examinations by the main and the second observer to assess intra- and interobserver reliabilities. (Line 134~136)

#35. General comment (Materials and Results): the statistical analyses section should be in standalone paragraph. Some fundamental criteria in chosen and reporting statistical analyses were ignored. For example, you test your data for normality distribution so as to be guide on which of the statistical test (parametric or non parametric) to be applied and also, to know which of the measure of central tendency (mean or median) or dispersion (standard deviation, inter quartile etc) to be used in summarizing your results. The acceptable level of significance and confidence level should be stated. The order and chronology of the statistical test should be from simple to complex. The multiple regressions should used to depict relationships.

- Response: We totally agree with and appreciate the reviewer’s opinion. We have reorganized the ‘materials and methods’ section in the order of ‘sample collection’, ‘estimation of dental age’ and ‘statistical analysis’ and in a stand-alone paragraph. The test for assumption of normality and the level of significance have been stated in the section. We could revise the drawbacks in the manuscript with your kind guidance. Thank you very much!

- Corrections: Intra- and interobserver reliabilities for the evaluation of maturity in M2s and M3s, were calculated using Cohen’s kappa statistics [23]. Before statistical analysis, the assumption of normality was checked based on the values of skewness and kurtosis and the Shapiro-Wilk test. All continuous variables had univariate normality, as shown by skew values < 2 and kurtosis values < 7. Descriptive analysis was presented as the mean and standard deviation for the developmental stage of teeth. Both estimated and chronological ages satisfied the normality assumption. We tested the Pearson correlation coefficients (PCCs) for observing correlations between the chronological and estimated ages. Agreement tests were assessed by Intraclass correlation coefficients (ICCs) by a two-way random-effects model with an absolute agreement. In addition, Bland-Altman plots were generated for a visual representation of estimated and chronological ages. To assess the 18-year threshold, sensitivity (the ratio of samples that were equal to or over the age threshold and were estimated to being equal to or over the age threshold), specificity (the ratio of samples that were truly under the age threshold and were estimated to being under the threshold), positive predictive value (PPV, the ratio of samples that were estimated to being equal to or over the age threshold and were truly equal to or over the age threshold), negative predictive value (NPV, the ratio of samples that were estimated to being under the threshold and were truly under the age threshold), and accuracy (the ratio of samples that were correctly estimated) were calculated. Population differences in tooth developmental status between Koreans and Japanese were tested using a two-tailed independent t-test. Linear regression was performed to model the relationship between treating the developmental stage of each tooth and chronological age as dependent variables; a residual plot was used to test the quality of the model. Regression was performed using two approaches, that is, treating the developmental stage of each tooth as discrete and continuous variables, as per Lee’s suggestions [4]. A two-tailed P-value < 0.05 was considered statistically significant. All statistical analyses were performed using SAS version 9.4 (SAS Institute, Cary, NC). (Line 146~170)

#36. Lines 143 and 148 “The results of the statistical analysis…” there is better way to present results. You should report the important components of the tables/figures the make reference to them.

- Response: We totally agree with and appreciate the reviewer’s opinion. In response to the reviewer’s comments, these sentences have been split and repositioned in their respective subsections with reorganization of the ‘results’ section. Thank you for your suggestion.

- Corrections: The box plots for the estimated ages with single tooth or combinations of two teeth were showed in S1 Fig. (Line 221~222)

The data using single molar, or two combined molars are presented in S2 Table. (Line 243~244)

The relationships between chronological age and developmental degrees of single tooth and two combined teeth are presented in S3 Table for Koreans and S4 Table for Japanese. (Line 264~266)

#37. Line 153, nothing like “excellent correlation” it is moderate or strong. Also, emphasize on the significant please.

- Response: We totally agree with and appreciate the reviewer’s opinion. In response to the reviewer’s comments, the word has been changed to ‘strong’, and P-value has been added in the end of the sentence in the ‘results’ section. With your kind opinion, we could revise the wrong expressions, especially the description of the statistical results. Thank you for your pointing it out. 

- Corrections: The PCCs and ICCs revealed a strong correlation between estimated and chronological ages in the Korean population (P < 0.001). (Line 231~232)

#38. Line 161, box plot for comparison or differences between group not really relationship plot.

- Response: We totally agree with and appreciate the reviewer’s opinion. In response to the reviewer’s comments, we have revised the title of Fig 2. Thank you for your suggestion.

- Corrections: Fig 2. Box plots for comparison between chronological and estimated ages. (Line 226)

#39. Lines 161-164, “The age was …female” is this part of the legend”? if yes please bold it. The same applies to lines 166 to 169.

- Response: Thank you for your suggestion. According to submission guidelines of Plos One, they have instructed to use bold for the figure titles only. I have attached the URL address for your reference. Thank you so much.

https://journals.plos.org/plosone/s/file?id=9cba/PLOS%20Manuscript%20Body%20Formatting%20Guidelines.pdf

#40. General comment (Results): You need to reorganize your results from simple to complex. For example you can start with descriptive statistics, comparison results, then the correlation and relationship results will be at the tail end. The results should also follow the order of the objectives of the study.

- Response: We totally agree with and appreciate the reviewer’s opinion. With your kind suggestions, the results section has become easier for the readers to understand. In response to the reviewer’s comments, the ‘results’ section had been reorganized in the order of ‘Datasets’, ‘Observer reliabilities’, ‘Descriptive statistics of M2s and M3s’, ‘Accuracy of estimated ages’, ‘Assessment of 18-year threshold’, and ‘Relationships between dental maturity and chronological ages’. Thank you very much!

- Corrections: Kindly refer to the revised manuscript.

#41. Please minimize the unnecessary horizontal line in your tables.

- Response: We totally agree with and appreciate the reviewer’s opinion. In response to the reviewer’s comments, we have re-designed the tables in the manuscript by eliminating unnecessary horizontal lines. Thank you so much.

- Corrections: Kindly refer to the revised manuscript.

#42. Tables of correlations and specificity, … lack associated P values.

- Response: Thank you for your kind comment. The P-value tells you whether the correlation coefficient is significantly different from 0. All results in Table 4 and Table 5 were P < 0.001. Therefore, we did not report the P-value of all point estimates (PCC, ICC, sen, spe, PPV, NPV, Acc) in Tables 4 and 5, instead we reported with the 95% confidence intervals. We have added a sentence that all data were P < 0.001 in the table legends. The tables were modified for indication of 95% CI. Thank you so much.

- Corrections: Kindly refer to the revised manuscript.

#43. Tables 5 and 6: the titles are inappropriate. Let the title depict the content of the tables. Avoid the words “statistical data”. The whole of the table need rearrangement, consider landscaping your tables.

- Response: We totally agree with and appreciate the reviewer’s opinion. In response to the reviewer’s comments, the titles were revised for depicting the contents of the tables. Thank you.

- Corrections: Table 2. Comparison of chronological age according to each stage of M2 between the Korean and Japanese populations. (Line 202-203)

Table 3. Comparison of chronological age according to each stage of M3 between the Korean and Japanese populations. (Line 208-209)

#44. Line 238, why revalidating?

- Response: Thank you for your kind point. This study was the first validation of Lee’s method. The word, ‘revalidation’, was a mistake. With your kind suggestion, we have revised the faulty expression in the sentence. Thank you for the comment.

- Corrections: The present study aimed at validating Lee’s age estimation method for confirmation of applicability when performing forensic practices and assessing the 18-year threshold based on the estimated age by Lee’s method in the Korean and Japanese populations. (Line 292-294)

#45. Lines 243 to 256 are part of results not discussion.

- Response: We totally agree with and appreciate the reviewer’s opinion. In response to the reviewer’s comments, we have re-organized the discussion section. Many sentences were moved to the results section, and were modified. There are a lot of changes, especially in the ‘discussion’ section. Thank you very much.

- Corrections: Kindly refer to the revised manuscript.

#46. General comment (Discussion): Most of the stuff here are results. The authors should lay more emphases on the implication of their findings. The scientific bases of the observed results need to also be discussed. Limitation and recommendation (based on the applicability of the study) can be stated at the tail end of this section.

- Response: We totally agree with and appreciate the reviewer’s opinion. In response to the reviewer’s comments, we have re-organized the discussion section. Many sentences were moved to results section, and some were modified and contracted. We have also added some sentences about emphasizing the findings, new references supporting our findings, limitation of this study, and necessity of future research. Thanks to your kind and dedicated review, our manuscript is much more systematic and logical. We greatly appreciate your effort and extend our sincere gratitude.

- Corrections: Discussion

 The present study aimed at validating Lee’s age estimation method for confirmation of applicability when performing forensic practices and assessing the 18-year threshold based on the estimated age by Lee’s method in the Korean and Japanese populations. The validity of Lee’s method was tested by calculating the correlation coefficients between estimated and chronologic ages, and the sensitivity and specificity were calculated by setting a threshold based on the age of 18 years for both populations. Further, we compared the differences in the chronological age according to the tooth development stage between the Korean and Japanese populations by regression analysis. (Line 292-299)

Based on the strong correlations (0.90 or more in PCCs) between estimated and chronologic ages in the Korean population, we confirmed the appropriateness of Lee et al.’s formula for estimating the age of the Korean population. Meanwhile, when Lee’s method was applied to the Japanese population, broader interquartile ranges of the estimated age (Fig 2, S1 Table), lower values of PCCs and ICCs (Table 4), and more scattering patterns in the Bland–Altman plot were observed (Fig 3). These results suggest that the accuracy of estimation may decrease if age estimation methods based on the Korean population are used for the Japanese population and imply possible population differences in dental development between the Korean and Japanese populations. (Line 308-316)

However, several studies have also been reported with the opposite results. Liversidge et al. [14, 40] examined whether ethnic differences existed in tooth maturity and reported that the accuracy of estimation was not vivid regardless of the specific reference from different population data. They asserted that the error of estimated ages which was reported by many population-specific researches, may be from the different sample distributions, and not from population differences. Rodriguez et al. [41] tested the validity of twelve age estimation methods which was based from non-mexican population data or international multi-population data in Mexican children. They reported quite good applicability of the tested methods and argued that the population differences in teeth development may be very small. Although the values of coefficients between the estimated and chronological ages in the Japanese population was lower in this study, it could also be evaluated as a strong correlation (0.80 to 0.72). In addition, we should consider that this study was analyzed with radiographs collected from a single institution from both countries. Therefore, generalization of possible population differences between the two populations based on the own results of this study to be looked at very carefully. If we want to confirm the results from this study and generalize the tendency to the entire population, future studies with data from multiple institutions should be performed. (Line 325-341)

The coefficients of determination (adjusted r2) between the stages of M2s and M3s and chronological age were lower in the Japanese population (0.596–0.693) than in the Korean population (0.834–0.855). Based on the results of regression analysis for the Japanese population, the coefficients of determination were decreased if the regression was performed with the combined developmental data of M3s and M2s, and the least values of adjusted r2 were observed on the regression with the combined developmental data of upper and lower M2s. (Lines 348-354)

---

## [Decision Letter · Decision Letter 1]

16 Jun 2022

PONE-D-22-06902R1Accuracy of age estimation and assessment of the 18-year threshold based on second and third molar maturity in Koreans and JapanesePLOS ONE

Dear Dr. Lee,

Thank you for submitting your manuscript to PLOS ONE. After careful consideration, we feel that it has merit but does not fully meet PLOS ONE’s publication criteria as it currently stands. Therefore, we invite you to submit a revised version of the manuscript that addresses the points raised during the review process.

We look forward to receiving your revised manuscript.

Kind regards,

Dinh-Toi Chu, PhD

Academic Editor

PLOS ONE

Journal Requirements:

Reviewers' comments:

Reviewer's Responses to Questions

**Comments to the Author**

1. If the authors have adequately addressed your comments raised in a previous round of review and you feel that this manuscript is now acceptable for publication, you may indicate that here to bypass the “Comments to the Author” section, enter your conflict of interest statement in the “Confidential to Editor” section, and submit your "Accept" recommendation.

Reviewer #1: All comments have been addressed

Reviewer #2: All comments have been addressed

Reviewer #3: All comments have been addressed

Reviewer #4: All comments have been addressed

2. Is the manuscript technically sound, and do the data support the conclusions?

Reviewer #1: Yes

Reviewer #2: Yes

Reviewer #3: Yes

Reviewer #4: Yes

3. Has the statistical analysis been performed appropriately and rigorously? 

Reviewer #1: Yes

Reviewer #2: Yes

Reviewer #3: Yes

Reviewer #4: Yes

4. Have the authors made all data underlying the findings in their manuscript fully available?

Reviewer #1: Yes

Reviewer #2: Yes

Reviewer #3: Yes

Reviewer #4: Yes

5. Is the manuscript presented in an intelligible fashion and written in standard English?

Reviewer #1: Yes

Reviewer #2: Yes

Reviewer #3: Yes

Reviewer #4: Yes

6. Review Comments to the Author

Reviewer #1: I kindly request the authors to look at their sample size calculation and sampling method, better if they consult statistician

I'm asking how do you reach into your final sample size? they said the minimum sample size was calculated as 403 but the final sample size was 2657, what is this miss much? How you withdraw your final samples from the source population?

Reviewer #2: (No Response)

Reviewer #3: Thank you so much; all of my concerns have been addressed. The study's findings are much too important for the field. As a result, I recommend publishing it as is.

Reviewer #4: There is significant improvement in the manuscript. All the comments have been well addressed. Now the work can be accepted for publication.

Best regards

7. PLOS authors have the option to publish the peer review history of their article (what does this mean?). If published, this will include your full peer review and any attached files.

Reviewer #1: No

Reviewer #2: **Yes: **Getachew Abebe

Reviewer #3: **Yes: **Fentahun Adane(PhD)

Reviewer #4: **Yes: **Dr Lawan Hassan Adamu

---

## [Author Response · Author response to Decision Letter 1]

17 Jun 2022

Respond to editor

#1. Please submit your revised manuscript by Jul 30 2022 11:59PM. If you will need more time than this to complete your revisions, please reply to this message or contact the journal office at plosone@plos.org. - Response: It is already done. Thank you.

#2. Please include the following items when submitting your revised manuscript:

- Response: All three files were checked and uploaded. Thank you.

#3. - Response: We also uploaded the updated “cover letter” which was indicating the changes in financial disclosure. Thank you so much. The underlined contents were added in financial disclosure.

- Corrections: This research was supported by Catholic Medical Center Research Foundation (SSL, 5-2021-B0001-00309) funded by The Catholic University of Korea and this work was also supported by National Forensic Service (SSL, NFS2022MED08), Ministry of the Interior and Safety, Republic of Korea. The funders had no role in study design, data collection and analysis, decision to publish, or preparation of the manuscript.

#4. Guidelines for resubmitting your figure files are available below the reviewer comments at the end of this letter.

- Response: There is no changes in figure files. Thank you. 

#5. If applicable, we recommend that you deposit your laboratory protocols in protocols.io to enhance the reproducibility of your results. Protocols.io assigns your protocol its own identifier (DOI) so that it can be cited independently in the future. For instructions see: https://journals.plos.org/plosone/s/submission-guidelines#loc-laboratory-protocols. Additionally, PLOS ONE offers an option for publishing peer-reviewed Lab Protocol articles, which describe protocols hosted on protocols.io. Read more information on sharing protocols at https://plos.org/protocols?utm_medium=editorial-email&utm_source=authorletters&utm_campaign=protocols.

- Response: There is no lab protocols in this study. Thank you.

#6. Please review your reference list to ensure that it is complete and correct. If you have cited papers that have been retracted, please include the rationale for doing so in the manuscript text or remove these references and replace them with relevant current references. Any changes to the reference list should be mentioned in the rebuttal letter that accompanies your revised manuscript. If you need to cite a retracted article, indicate the article’s retracted status in the References list and also include a citation and full reference for the retraction notice.

- Response: We checked the reference list again, there was no retracted article, and we confirmed the reference list in this manuscript was complete and correct. Thank you so much.

Respond to reviewer

Reviewer: 1 

#1. I kindly request the authors to look at their sample size calculation and sampling method, better if they consult statistician. I'm asking how do you reach into your final sample size? they said the minimum sample size was calculated as 403 but the final sample size was 2657, what is this miss much? How you withdraw your final samples from the source population?

- Response: With the opinion from reviewer #1, we checked our manuscript again, and found the expression might be rise the misunderstanding. The sample size was calculated as 403 for the single regression model. In this study, the regression model was performed on a population stratified by gender and nation. Therefore, the sample size which we presented was over 403. In our opinion, we used only the minimum sample size for each regression model and the total samples we analyzed was 2657. Nevertheless, we corrected sentences that may cause misunderstandings by the readers. Thank you for your suggestion. It can make this manuscript clearer for readers.

- Corrections: Data were randomly selected and stratified by gender and population from the radiographs which were enrolled from Seoul St. Mary’s Hospital, Catholic University of Korea (900 males and 900 females, 19.47±2.62 years), and Iwate Medical University of Japan (406 males and 451 females, 19.31±2.60 years). (Line 101~104)

---

## [Editor Report · Decision Letter 2]

27 Jun 2022

Accuracy of age estimation and assessment of the 18-year threshold based on second and third molar maturity in Koreans and Japanese

PONE-D-22-06902R2

Dear Dr. Lee,

We’re pleased to inform you that your manuscript has been judged scientifically suitable for publication and will be formally accepted for publication once it meets all outstanding technical requirements.

Kind regards,

Dinh-Toi Chu, PhD

Academic Editor

PLOS ONE

---

## [Editor Report · Acceptance letter]

29 Jun 2022

PONE-D-22-06902R2 

Accuracy of age estimation and assessment of the 18-year threshold based on second and third molar maturity in Koreans and Japanese 

Dear Dr. Lee:

I'm pleased to inform you that your manuscript has been deemed suitable for publication in PLOS ONE. Congratulations! Your manuscript is now with our production department. 

Kind regards, 

on behalf of

Dr. Dinh-Toi Chu 

Academic Editor

PLOS ONE